# Human Gingival Crevicular Fluids (GCF) Proteomics: An Overview

**DOI:** 10.3390/dj5010012

**Published:** 2017-02-22

**Authors:** Zohaib Khurshid, Maria Mali, Mustafa Naseem, Shariq Najeeb, Muhammad Sohail Zafar

**Affiliations:** 1Prosthodontics and Implantology, College of Dentistry, King Faisal University, Al-Ahsa 31982, Saudi Arabia; 2Department of Orthodontics, Fatima Jinnah Dental College, Karachi 78650, Pakistan; drmariahmali@gmail.com; 3Preventive Dental Sciences, College of Dentistry, Dar-Al-Uloom University, Riyadh 13314, Saudi Arabia; m.naseem@dau.edu.sa; 4Department of Dentistry, Riyadh Consultative Clinics, Riyadh 11313, Saudi Arabia; shariqnajeeb@gmail.com; 5Department of Restorative Dentistry, College of Dentistry, Al-Taibah University, Medina Munawwarah 41311, Saudi Arabia; MzAFAR@taibahu.edu.sa; 6Department of Dental Materials, Islamic International Dental College, Riphah International University, Islamabad 44000, Pakistan

**Keywords:** proteomics, proteins, biomarkers, gingival crevicular fluids (GCFs), dentistry

## Abstract

Like other fluids of the human body, a gingival crevicular fluid (GCF) contains proteins, a diverse population of cells, desquamated epithelial cells, and bacteria from adjacent plaque. Proteomic tools have revolutionized the characterization of proteins and peptides and the detection of early disease changes in the human body. Gingival crevicular fluids (GCFs) are a very specific oral cavity fluid that represents periodontal health. Due to their non-invasive sampling, they have attracted proteome research and are used as diagnostic fluids for periodontal diseases and drug analysis. The aim of this review is to explore the proteomic science of gingival crevicular fluids (GCFs), their physiology, and their role in disease detection.

## 1. Introduction

Proteomic science aids in different areas of biomedical and clinical sciences with tools such as polyacrylamide gel electrophoresis (PAGE), high-pressure liquid chromatography (HPLC), mass spectrometry (MS), matrix-assisted laser desorption ionization (MALDI), and surface-enhanced laser desorption/ionization mass spectrometry (SELDI-MS) [1]. This science provides an understanding of healthy and diseased states of the human body through protein nature. Protein builds the human body and is known as the “working horses” of a cell. The ‘proteome’ is the cell’s proteins content. In short, proteomics is the discovery technology that rescues the understanding of the molecular behaviour of protein activity and how they respond to the process of a disease. Human body fluids such as blood, cerebrospinal fluid (CSF), saliva, gingival crevicular fluid (GCF), sera, urine, vaginal secretion, breast milk, sputum, peritoneal fluid, pleural fluid, and pericardial fluids are approved clinical samples for the diagnosis and maintenance of a disease state [2].

Human gingival crevicular fluid (GCF) was discovered in the nineteenth century, and its composition and oral defense mechanism were demonstrated by Brill and Björn in 1959 [3]. GCF is a physiological fluid that is classified as inflammatory exudate by many investigators, and some suggest it is an altered tissue transudate in a normal healthy state (see Figure 1). Originally, they originate from the gingival plexus of blood vessels in the gingival corium and are subjacent to the epithelium lining of the dental–gingival space. Its composition and flow were demonstrated by Waerhaug in 1952 [4,5]. Later, in 1974, Alfano presented two mechanisms based on GCF origination that include the engendering of standing osmotic gradient and the induction of classical inflammation [6]. GCF is used to detect periodontal diseases such as gingivitis, periodontitis (chronic and aggressive), and drug presence in periodontal pockets through the systematic pathway and is currently extensively used for proteomic analysis [7].

GCF basically contains local breakdown products such as tissues, inflammatory mediators, host inflammatory mediators, serum transudate (found in gingival sulcus), subgingival microbial plaque, extracellular proteins, and cells [8]. In Figure 2, detail of GCF composition is illustrated. This composition varies between periodontium in healthy and diseased conditions [9]. The amount of GCF production is quite small and varies according to the size of the gingival sulcus. Few investigators have measured GCF, and all observations have differed due to the variation in the GCF samples collected.

One group reported that the mean GCF volume ranged from 0.43 to 1.56 μL in the proximal spaces from the molar teeth [10]. Another group collected GCF from sulcus of slightly inflamed gingiva and reported approximately 0.1 mg in 3 min [11], which is the amount of GCF affected by mechanical factors, habitats (tobacco, smoke, and shisha), circadian periodicity, sex hormones, and periodontal surgeries (see Table 1). Engelberg et al. experimented on dogs by inducing a carbon gelatine mixture into the blood stream to understand the organization of vessels at the dento–gingival junction. Hence, it was histologically postulated that, in healthy tissue, these particles could not enter the intracellular spaces and remained in the capillaries. It was further shown that, in a state of acute inflammation, these particles can be found; in a healthy state, the vessels are organized in layers and close to gingival crevicular epithelium, whereas, in diseased or inflammation states, the layers are replaced by a loop form [12]. Furthermore, a series of experiments was performed, and it was concluded that capillary permeability varied according to the stimulus given and significantly increased in an inflamed state. This capillary permeability occasionally responded in a healthy state while chemo-mechanical massaging of the gingivae also showed changes. It was noted that there was a remarkable rise in pH of gingival sulcus as periodontitis developed, marked around 8.5. The rise in pH is associated with the destruction of proteins by bacteria in the sulcus. Ammonium that is basic in nature is produced as a by-product after the degradation of proteins and is thought to be the reason for the increase in pH of the crevicular sulcus. Other factors that are found to play a role in the progression from a healthy to a diseased state includes the oxygen level, the temperature, the redox potential, and the osmotic pressure [9].

Nature has provided the periodontium with connective tissue that is enriched with cellular and molecular components of blood; hence, gingival sulcus is consistently bathed by GCF [13]. This crevicular fluid provide a channel for the transportation of bacterial by-products into periodontium, as well as help to drive off host-derived by-products [14]. Many methods are available for the collection of GCF such as Intrasulcular and extrasulcular (see Figure 3) with the help of paper strips, capillary tubes, micropipette, gingival wash, and paper cones [5]. Other methods include using platinum loops, plastic strips, and paper points. In the last decade, researchers have favored using the paper strip in their research work due to easy insertion into a gingival crevice up to 1 mm of depth without bleeding from periodontal pockets [15] .

## 2. GCF as a Diagnostic Tool for Analysis of Oral Diseases

The oral cavity is a reservoir of the microbiome and harbours them, when unfavourable changes occur within the oral cavity; it results in pathological changes, such as gingivitis, periodontitis, and dental caries [16,17]. GCF, as a biomonitoring fluid, plays a constructive role in the diagnosis of oral diseases, especially for periodontitis and gingivitis. Its limited amount compromises the biochemical and proteomic analysis, and the severity of inflammation in periodontium affects its collection [15]. The introduction of mass spectrometry, with highly sensitive techniques, helping in the detection of protein and their components within many biological samples [18]. Its non-invasive collection technique helps in a sampling of any age group of human subjects and the attraction of allowing multiple sites for the sampling within the oral cavity. Until now, various inflammatory factors have been isolated from GCF, including cytokines, phosphatase, proteinase, local tissue degradation products, and proteins [19]. These factors have been reported to be possible diagnostic markers in periodontitis. GCF not only can be a future diagnostic tool in the identification of periodontitis but also can aid in detecting the progression of this disease. Early detection of periodontitis progression can be clinically useful by providing better control of disease activity and can improve patient monitoring [20]. Therefore, collecting GCF from multiple sites can aid in identifying high susceptibility for disease activity on those sites, the targeted risk site can be managed properly, and further progression of the disease can be controlled [21]. Huynh et al. studied the proteomic composition of GCF in healthy periodontium and compared them with proteome present in GCF of patients with gingivitis and chronic periodontitis. The GCF proteome analyzed was found different in each clinical condition and hence suggests that this analysis can assist us in knowing pathogenesis of the periodontal disease [22]. Tsuchida et al. collected GCFs from a healthy patient and compared them with patients having mild to moderate periodontal disease and with a patient suffering from the severe periodontal disease. The gel-free proteomic approach was used, and GCFs were labeled by a reagent called as Tandem Mass Tags (TMTs). In this method, 619 proteins were analyzed from the above-mentioned samples. Their reports showed, in severe periodontitis, higher amounts of lipocalin2 (LCN2) and matrix metalloproteinase-9 (MMP-9) were expressed. Furthermore, Pisano et al. worked on the peptides of GCF by using HPLC-ESI-MS and detected a high concentration of α-defensins, and small amounts of cystatin-A, basic PB peptides, and statherin were also detected [20].

External root resorption is a universal phenomenon that is associated with many factors such as trauma, orthodontic tooth movement periodontal disease, and physiological shedding of deciduous teeth. Rody Jr et al. worked to identify the biomarkers in GCF by using LC-MS (liquid chromatography-mass spectrometry). He postulated 2789 proteins in a control group, and 2421 proteins in resorption samples were identified. Therefore, these biomarkers can provide us with a new possible tool for the early detection of external root resorption so that orthodontic tooth movement should avoided, and other conditions mentioned above will better be managed [23].

Orthodontics is one of the major subspecialties of dentistry that deals with the management of malposition teeth and the jaws. Certain studies were conducted to assess changes within GCF during orthodontic tooth movement [24]. Initially, when orthodontic force is applied to allow tooth movement to occur, certain metabolic changes occur within periodontium along with the process of bone remodeling that includes some osteoblastic and osteoclastic activity. This biological and physiological process results in acute inflammatory response that occurs within periodontal space. Lactate dehydrogenase, an important enzyme that is released out after cell death, was reported as a potential marker, as its level was found to be elevated within GCF during orthodontic treatment. Serra et al. demonstrated the LDH level in GCF during the early stage of orthodontic treatment and formulated that, when orthodontic force is applied, the periodontium either goes under tension or compression, resulting in cell death. As hypothesized, the LDH level in GCF was elevated in those sites enduring orthodontic force and was independent of age and sex [25].

Bone cells are deposited at the site of tension and resorbed at the site of compression. A longitudinal study was performed to assess the level of alkaline phosphatase within GCF in a patient undergoing orthodontic treatment [26]. It was noted that the enzyme level was elevated at the site of tension, as compared to the site of compression. Hence, these enzymes represent future novel markers and are associated with gingival inflammation caused during forces applied by orthodontic appliances. Orthodontics demands accurate timings for the management of most skeletal discrepancies; different stages of bone growth and its maturation in the subject must be known. This is usually achieved most commonly by identifying and comparing chronological age and dental age, and by CVM staging and hand-wrist radiograph. The former methods are relatively unreliable, and the latter methods involve radiation exposure [27]. Hence, any method that is noninvasive with minimum side effects and that can help the clinician to identify the growth spurt is in highly demand. Research that showed variable peptides in the GCF of both pre-pubertal and post-pubertal subjects has been conducted recently [28]. Further studies are encouraged to find more information about these novel biomarkers.

## 3. GCF Proteomic Analysis

After collection, GCF samples will go through different steps of proteomics analysis, as illustrated in Figure 4. A variety of proteolytic enzymes has been identified in GCF, such as collagenase, elastase, and cathepsin B, D, H, and L [29]. These proteolytic enzymes are reported as the destructor of periodontal tissues and have the capability of degrading type-I collagen and glycoproteins [30].

The most commonly reported identified proteins from GCF are actin, keratins, histones, annexins, proteins S100-A9, apolipoprotein A-1, albumin, salivary gland antimicrobial peptides (histatins, HNP-1, -2 & -3, LL-37, statherin), and cystatin B [31,32]. Some immune-related proteins present in GCF include Ig gamma-1 chain C region, Ig gamma-3 chain C region, lactoferroxin-C, leukocyte elastase inhibitor, alpha 1 antitrypsin, heat shock protein beta-1, and coronin-1A [33]. Table 2 presents a detailed profiling of GCF proteins, proteomic tools used, and some of the proteins identified.

## 4. Conclusions

Gingival crevicular fluid is a serum exudate that originates from the periodontal sulcus or pocket and is regarded as a promising biological fluid for the detection of periodontal disease. Its composition resembles normal serum, but its volume fluctuates in certain conditions such as those of gingivitis, caries, external root resorption, and chronic periodontitis, as well as during orthodontic tooth movement. GCF is composed of variable substances that include immunoglobulin, enzymes, local mediators, toxin cells, protein peptides, tissue breakdown products, and microorganisms. The level of this substance when fluctuating in the above-mentioned conditions, as reported in many papers, will mark as a future diagnostic tool in their non-invasive analysis. Due to limitations in its collection, which includes volume size and contamination, collecting methods need further work, and a way to improve the ease for clinicians must be found; such development would help us to better demonstrate the pathogenesis of such diseases and to determine better strategies for treatment and early prevention.

## Figures and Tables

**Figure 1 dentistry-05-00012-f001:**
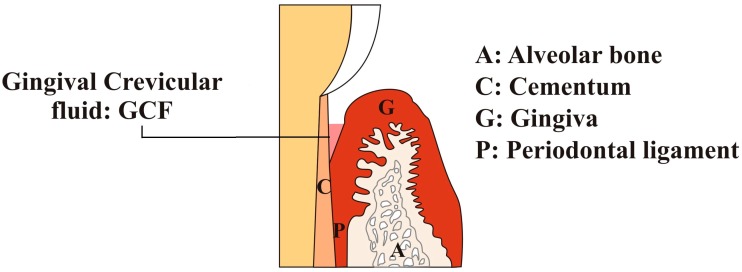
The anatomical location of gingival crevicular fluid (GCF) in a healthy subject.

**Figure 2 dentistry-05-00012-f002:**
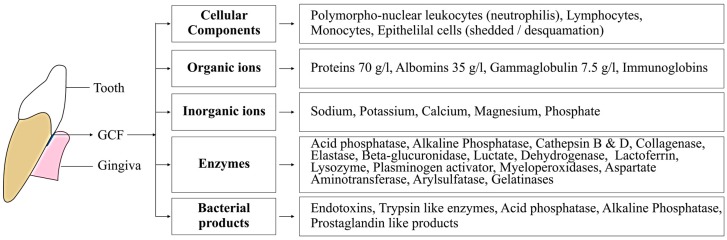
Illustration describing the composition of gingival crevicular fluids (GCFs).

**Figure 3 dentistry-05-00012-f003:**
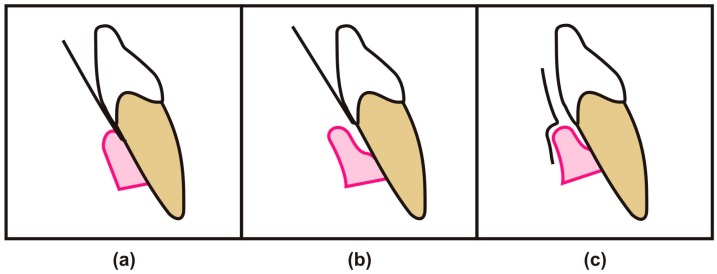
Illustration representing the different approaches of collecting gingival crevicular fluids (GCFs) from the oral cavity. (**a**) Intrasulcular approach; (**b**,**c**) Extrasulcular approach.

**Figure 4 dentistry-05-00012-f004:**
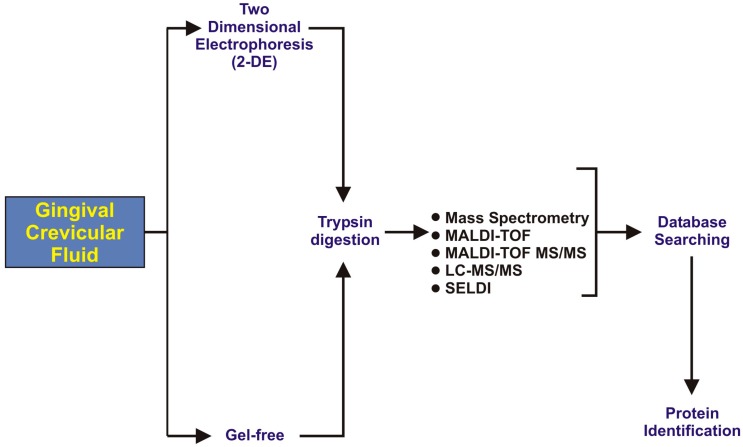
A flow diagram representing the proteomic analysis of healthy and diseased conditions via gingival crevicular fluid (GCF), adapted from Khurshid et al. [16].

**Table 1 dentistry-05-00012-t001:** Description of factors affecting the amount of GCF in the human oral cavity.

Factors	Description
Mechanical	Chewing coarse foods, vigorous brushing and gingival massage are known to increase GCF production
Circadian periodicity	The amount of GCF increases gradually from 6 a.m. to 10 p.m. and it decreases after that
Periodontal surgeries	GCF production increases after periodontal surgeries, during the healing period
Smoking	Smoking increases GCF flow. This increase in GCF due to smoking is immediate and transient

**Table 2 dentistry-05-00012-t002:** Description of reported studies on GCF biomarkers analysis.

Author & Year	Sample Collection Sites	Collection Method	Proteomic Tool	Number of Identified Proteins	Outcome of Study	Ref.
Baliban et al., 2012	Collected from pre-selected sites with probing depth >6 mm and <8 mm in periodontitis patients and for periodontal health from mesio-buccal sites of first molar	Filter strips (Periopapers^®^, Interstate Drug Exchange, Amityville, NY, USA)	Protein digest with trypsin, HPLC, fragmented analysis with tandem mass spectrometry (MS/MS)	432 human proteins identified (120 new)	Studied identified novel biomarkers from GCF of periodontal healthy and chronic periodontitis patients	[31]
S.Tsuchida et al., 2012	Labial side of maxillary incisors without crown and restoration	Absorbent paper points (ZIPPERER^®^, Munich, Germany)	2DE, SDS-PAGE, Western Blot analysis, HPLC with LTQ-XL, HPLC with LTQ-Orbitrap XL, LC-MS/MS	327 proteins identified	SOD1 and DCD were significantly ↑ in GCF of periodontal patients	[14]
Carneiro et al., 2012	Healthy gingival sulcus of the second and third molar teeth	Periopapers^®^, USA	Trypsin digested followed by nano-flow liquid chromatography electrospray ionization tandem mass spectrometry (LC-ESI-MS/MS) analysis and enzyme-linked immunosorbent essay (ELISA) for human albumin analysis	199 proteins identified	Provide protein analysis of healthy periodontium and explore GCF composition with new groups of proteins specific to GCF microenvironment	[18]
Ngo et al., 2013	Five deepest sites and molar sites except mesial surface were excluded	Microcaps (glass micocapillary tubes); Drummed Scientific, Brookmall, PA, USA	Matrix-assisted laser desposition/ionization time-of-flight (MALDI-TOF) mass spectrometry	-	GCF mass spectra could be best for analyzing attachment loss and diagnosis of periodontal diseases	[32]
Carina M., et al. 2013	Chronic Periodontitis patients sample were taken from different sites (5 deep sites, 5 shallow sites with gingivitis, and 4 without bleeding on probing sites)	Periopaper strip (ProFlow Inc. Amityville, NY, USA)	Reversed-phase LC-ESi-MS/MS and ELISA	230 proteins identified	Concluded marked differences in GCF proteomics in different disease profiles	[33]
Carneiro et al., 2014	The pre-selected specific sites with moderate and severe chronic periodontal disease were defined by pocket depth of 5–7 mm (24 patients) and >7 mm (16 patients)	Periopaper strips (Oraflow, Plainview, NY, USA)	SDS-PAGE, Isotope-Coded-Affinity-Tag (ICAT) labeling, mTRAQ labeling, Nano-LC-ESI-MS/MS, Human Albumin ELISA Kit, and S100-A9 protein quantification by ELISA	199 proteins Identified	Innovative approach concluded the novel changes in host and microbial derived GCF proteome of periodontal patients	[19]
Rody, Jr. et al., 2014	Collected from a deciduous second molar with radiographic evidence of root resorption on 1 quadrant (experimental site) and from the permanent first molar on the contralateral quadrant (control site) in the same jaw.	Periopaper strips (Oraflow, Plainview, NY, USA)	One dimensional LC-MS and Two dimensional LC-MS	2789 proteins in control group and 2421 proteins in root resorption group	Mass spectrometry is a useful tool for analyzing external root resorption	[23]
Kinney et al., 2014	Collection from the mesio-buccal aspect of each site (tooth) for up-to 28 teeth per patient.	Methylcellulose strip (Pro Flow, Inc., Amityville, NY, USA)	ELISA and Quantibody Human Cytokine Array	-	This method offers improved patient monitoring and disease control	[21]
Huynh et al., 2015	Collection were chosen based on how well they represented the healthy, gingivitis, and chronic periodontitis inclusion criteria	Glass-microcapillary tube (Drummond Scientific, Brookmall, PA, USA)	One-dimensional Gel Electrophoresis and Nano-LC-ESI-MS	121 proteins identified	Concluded various biomarkers which differentiate between healthy periodontium, gingivitis, and chronic periodontitis	[22]

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
