# Peer review of "Human Gingival Crevicular Fluids (GCF) Proteomics: An Overview"

_dentistry, 2017, doi:10.3390/dj5010012_

Round 1
Reviewer 1 Report
The paper is clearly written and can be published ad it is
Author Response
Thanks for appreciating this manuscript for publishing. This fluid will revolutionize the diagnostics sciences field.
Reviewer 2 Report
This review describes 1) general features of gingival crevicular fluids (GCF) and sample collection methods, 2) proteomic analysis of GCF for diagnostics, and 3) proteomic procedures. This review would help understanding the current status and prospects of biomarker discovery from GCF. There are several points to be considered (shown below).
1) One of the critical points in current proteomic research is how to quantitate proteins.It is especially important in exploring biomarkers using LC-MSMS techniques. However, any general or specific comments on quantitation were not given in the manuscript.
2) Because Section 3 shows the experimental procedures of proteomics analysis, while Section 2 shows the results from proteomic analyses, it seems more proper to place Section 3 before Section 2.
3) It is difficult to for me parse the following sentences.
Lines 21-23, “increasing interest….”
Lines 110-112, “by collecting…”.
Lines 118-119, “in this gel-free….”
Author Response
Dear Reviewer
First of all, we all are thankful to you for giving time to our effort. We do our level best for the improvement and accept all the suggestion suggested by you.
Comment-1;
Regarding quantification of proteins in GCF is well highlighted by the reviewer but at this moment a group of the researcher trying to analysing it's important first, and then standardisation will be established for sampling like the blood sample.
Comment-2
We can't move section 3 before section 2 because in section two we detailed discussed the importance and its sampling method see figure- 3. Before sampling, if we shift it in the place of part 2 look bad for readers.
Comment-3
we fixed all lines see track changes in the main file. Thanks
Reviewer 3 Report
Very interesting and well written paper. Useful to readers
Author Response
Thanks for the appreciating comments.
